# Adaptive evolution shapes the present-day distribution of the thermal sensitivity of population growth rate

Dimitrios—Georgios Kontopoulos[1,2]*, Thomas P. Smith[2], Timothy G. Barraclough[2,3], Samraat Pawar[2]

1 Science and Solutions for a Changing Planet DTP, Imperial College London, London, United Kingdom,
2 Department of Life Sciences, Imperial College London, Silwood Park, Ascot, Berkshire, United Kingdom,
3 Department of Zoology, University of Oxford, Oxford, Oxfordshire, United Kingdom

* dgkontopoulos@gmail.com

**Data Availability Statement:** Data and source code for the analyses of the present study are available at https://doi.org/10.6084/m9.figshare.12816140.v1 and https://github.com/

## Abstract

Developing a thorough understanding of how ectotherm physiology adapts to different thermal environments is of crucial importance, especially in the face of global climate change. A key aspect of an organism's thermal performance curve (TPC)—the relationship between fitness-related trait performance and temperature—is its thermal sensitivity, i.e., the rate at which trait values increase with temperature within its typically experienced thermal range. For a given trait, the distribution of thermal sensitivities across species, often quantified as "activation energy" values, is typically right-skewed. Currently, the mechanisms that generate this distribution are unclear, with considerable debate about the role of thermodynamic constraints versus adaptive evolution. Here, using a phylogenetic comparative approach, we study the evolution of the thermal sensitivity of population growth rate across phytoplankton (Cyanobacteria and eukaryotic microalgae) and prokaryotes (bacteria and archaea), 2 microbial groups that play a major role in the global carbon cycle. We find that thermal sensitivity across these groups is moderately phylogenetically heritable, and that its distribution is shaped by repeated evolutionary convergence throughout its parameter space. More precisely, we detect bursts of adaptive evolution in thermal sensitivity, increasing the amount of overlap among its distributions in different clades. We obtain qualitatively similar results from evolutionary analyses of the thermal sensitivities of 2 physiological rates underlying growth rate: net photosynthesis and respiration of plants. Furthermore, we find that these episodes of evolutionary convergence are consistent with 2 opposing forces: decrease in thermal sensitivity due to environmental fluctuations and increase due to adaptation to stable environments. Overall, our results indicate that adaptation can lead to large and relatively rapid shifts in thermal sensitivity, especially in microbes for which rapid evolution can occur at short timescales. Thus, more attention needs to be paid to elucidating the implications of rapid evolution in organismal thermal sensitivity for ecosystem functioning.

dgkontopoulos/Kontopoulos_et_al_thermal_sensitivity_2020, respectively.

**Funding:** DGK was supported by a Natural Environment Research Council (NERC) Doctoral Training Partnership (DTP) scholarship (NE/L002515/1; http://gotw.nerc.ac.uk/list_full.asp?pcode=NE%2FL002515%2F1). TPS was supported by a Biotechnology and Biological Sciences Research Council (BBSRC) DTP scholarship (BB/J014575/1; https://bbsrc.ukri.org/research/grants-search/AwardDetails/?FundingReference=BB%2FJ014575%2F1). SP was supported by NERC grants NE/M004740/1 (http://gotw.nerc.ac.uk/list_full.asp?pcode=NE%2FM004740%2F1) and NE/M020843/1 (http://gotw.nerc.ac.uk/list_full.asp?pcode=NE%2FM020843%2F1). The funders had no role in study design, data collection and analysis, decision to publish, or preparation of the manuscript.

**Competing interests:** The authors have declared that no competing interests exist.

**Abbreviations:** DIC, deviance information criterion; HPD, highest posterior density; MTE, Metabolic Theory of Ecology; TPC, thermal performance curve; UTD, universal temperature dependence.

## Introduction

Current climate change projections suggest that the average global temperature in 2100 will be higher than the average of 1986–2005 by 0.3 °C–4.8 °C [1], coupled with an increase in temperature fluctuations in certain areas [2]. Therefore, it is now more important than ever to understand how temperature changes affect biological systems, from individuals to whole ecosystems. At the level of individual organisms, temperature affects functional traits in the form of the thermal performance curve (TPC). Typically, this TPC, especially when the trait is a rate (e.g., respiration rate, photosynthesis, growth), takes the shape of a negatively skewed unimodal curve (Fig 1) [3, 4]. The curve increases (approximately) exponentially to a maximum ($T_{pk}$) and then also decreases exponentially, with the fall being steeper than the rise. Understanding how various aspects of the shape of this TPC adapt to a changing thermal environment is crucial for predicting how rapidly organisms can respond to climate change.

According to the Metabolic Theory of Ecology (MTE) as well as a large body of physiological research, the shape of the TPC is expected to reflect the effects of temperature on the kinetics of a single rate-limiting enzyme involved in key metabolic reactions [5, 8–11]. Under this assumption, the rise in trait values up to $T_{pk}$ can be mechanistically described using the Boltzmann-Arrhenius equation:

$$B(T) = B_0 \cdot e^{\left[\frac{-E}{k}\left(\frac{1}{T} - \frac{1}{T_{ref}}\right)\right]}. \tag{1}$$

Here, $B$ is the value of a biological trait, $B_0$ is a normalisation constant—that includes the effect of cell or body size—which gives the trait value at a reference temperature ($T_{ref}$), $T$ is temperature (in K), $k$ is the Boltzmann constant ($8.617 \cdot 10^{-5}$ eV $\cdot K^{-1}$), and $E$ (eV) is the thermal sensitivity of the trait at the rising component of the TPC up to $T_{pk}$. Because $T_{pk}$ tends to be higher than the mean environmental temperature [6, 7, 12], $E$ represents the thermal sensitivity within the organism's typically experienced thermal range.

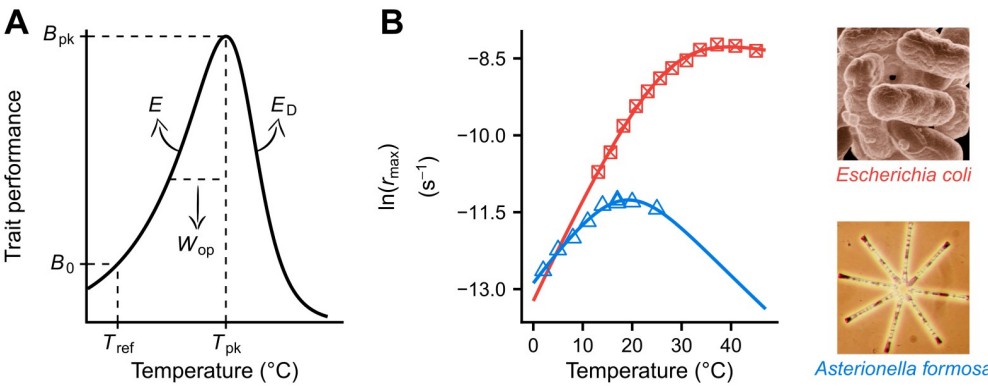

**Fig 1. The TPC of ectotherm metabolic traits, as described by the Sharpe-Schoolfield model [5].** (A) $T_{pk}$ (K) is the temperature at which the curve peaks, reaching a maximum height that is equal to $B_{pk}$ (in units of trait performance). $E$ and $E_D$ (eV) control how steeply the TPC rises and falls, respectively. $B_0$ (in units of trait performance) is the trait performance normalised at a reference temperature ($T_{ref}$) below the peak. In addition, $W_{op}$ (K), the operational niche width of the TPC, can also be calculated a posteriori as the difference between $T_{pk}$ and the temperature at the rise of the TPC where $B(T) = 0.5 \cdot B_{pk}$. We note that we use $W_{op}$ instead of a metric that captures the entire TPC width because previous studies have shown that species generally experience temperatures below $T_{pk}$ [6, 7]. Thus, $W_{op}$ is a measure of the thermal sensitivity of the trait near typically experienced temperatures. (B) TPCs of individual- and population-level traits (such as $r_{max}$) are usually well described by the Sharpe-Schoolfield model. The raw data for panel B are available at https://doi.org/10.6084/m9.figshare.12816140.v1. TPC, thermal performance curve.

Early MTE studies argued that, because of strong thermodynamic constraints, adaptation will predominantly result in changes in $B_0$, whereas $E$ will remain almost constant across traits (e.g., respiration rate, $r_{max}$), species, and environments around a range of 0.6–0.7 eV [8–10]. The latter assumption is referred to in the literature as universal temperature dependence (UTD). This restricted range of values that $E$ can take is centered on the putative mean activation energy of respiration ($\approx 0.65$ eV). A notable exception to the UTD is photosynthesis rate, which is expected to have a lower $E$ value of $\approx 0.32$ eV, reflecting the activation energy of the rate-limiting steps of photosynthesis [13].

The existence of a UTD has been strongly debated. From a theoretical standpoint, critics of the UTD have argued that the Boltzmann-Arrhenius model is too simple to mechanistically describe the complex physiological mechanisms of diverse organisms [3, 14–16] and is inadequate for describing TPCs emerging from the interaction of multiple factors, and not just the effects of temperature on enzyme kinetics. That is, the $E$ calculated by fitting the Boltzmann-Arrhenius model to biological traits is an emergent property that does not directly reflect the activation energy of a single rate-limiting enzyme. For example, a fixed thermal sensitivity for net photosynthesis rate is not realistic because it depends on the rate of gross photosynthesis as well as photorespiration, which is in turn determined not only by temperature but also by the availability of $CO_2$ in relation to $O_2$ [17].

Indeed, there is now overwhelming empirical evidence for variation in $E$ (thermal sensitivity) far exceeding the narrow range of 0.6–0.7 eV, with such variation being, to an extent, taxonomically structured [12, 18–23]. Furthermore, the distribution of $E$ values across species is typically not Gaussian but right-skewed. If we assume that $E$ is nearly constant across species —and therefore that variation in $E$ is mainly due to measurement error—such skewness could be the outcome of the proximity of the $E$ distribution to its lower boundary (0 eV). In that case, however, we would expect a high density of $E$ values close to 0 eV, but such a pattern has not been observed [18]. Both the deviations from the MTE expectation of a heavily restricted range for $E$ and the shape of its distribution have been argued to be partly driven by adaptation to local environmental factors by multiple studies. These include selection on prey to have lower thermal sensitivity than predators (the "thermal life-dinner principle") [18], adaptation to temperature fluctuations within and/or across generations [3, 21, 24–26], and adaptive increases in carbon allocation or use efficiency due to warming [27–30].

In general then, adaptive changes in the TPCs of underlying (fitness-related) traits are expected to influence the TPCs of higher-order traits such as $r_{max}$, resulting in deviations from a UTD. Therefore, understanding how the thermal sensitivity of $r_{max}$ and its distribution evolves is particularly important, as it may also yield useful insights about the evolution of the TPCs of underlying physiological traits (e.g., respiration rate, photosynthesis rate, and carbon allocation efficiency). Indeed, systematic shifts in the thermal sensitivity of fundamental physiological traits have been documented [27, 31–33], albeit not through comparative analyses of large datasets.

In particular, phylogenetic heritability—the extent to which closely related species have more similar trait values than species chosen at random—can provide key insights regarding the evolution of thermal sensitivity. A phylogenetic heritability of 1 indicates that the evolution of the trait across the phylogenetic tree is indistinguishable from a random walk (Brownian motion) in the parameter space. Note that this does not necessarily indicate that the trait evolves neutrally, as it may be under selection towards a nonstationary optimum that itself performs a random walk [34]. In contrast, a phylogenetic heritability of 0 indicates that trait values are independent of the phylogeny. This is the case either because (i) the trait is practically invariant across species and any variation is due to measurement error, or (ii) the evolution of the trait is very fast and with frequent convergence (i.e., independent evolution of similar trait

values by different lineages). It is worth clarifying that rapid trait evolution that does not result in convergence (e.g., when major clades are extremely separated in the parameter space) will not lead to a complete absence of phylogenetic heritability. Phylogenetic heritabilities between 0 and 1 reflect deviations from Brownian motion (e.g., due to occasional patterns of evolutionary convergence). Among phytoplankton, measures of thermal sensitivity of $r_{max}$ ($E$ and $W_{op}$) have previously been shown to exhibit intermediate phylogenetic heritability [35]. This indicates that, among phytoplankton, thermal sensitivity is not constant but evolves along the phylogeny, albeit not as a purely random walk in trait space, reflecting either thermodynamically constrained evolution or rapid evolution in response to selection.

To understand (i) how variation in thermal sensitivity accumulates across multiple autotroph and heterotroph groups and (ii) whether its distribution is shaped by environmental selection, here we conduct a thorough investigation of the evolutionary patterns of thermal sensitivity, focusing particularly on $r_{max}$. Using a phylogenetic comparative approach, we test the following hypotheses:

## 1. Thermal sensitivity does not evolve across species and any variation is noise-like

In this scenario, thermodynamic constraints would force $E$ to be distributed around a mean of 0.65 eV (or 0.32 eV in the case of photosynthesis), with deviations from the mean being mostly due to measurement error. Depending on the magnitude of the error, the $E$ distribution would either be approximately Gaussian (little measurement error) or non-Gaussian with a high density near 0 eV (substantial measurement error). This hypothesis agrees with the UTD concept of early MTE studies. If this hypothesis holds, thermal sensitivity would have 0 phylogenetic heritability and would not vary systematically across different environments.

## 2. Thermal sensitivity evolves gradually across species but tends to revert to a key central value, without ever moving very far from it

This hypothesis is also consistent with the UTD assumption, as it is a relaxed variant of hypothesis 1. Here, small deviations from the central tendency of 0.65 eV (or 0.32 eV) are possible, as they would reflect adaptation of species' enzymes to certain ecological lifestyles or niches. Therefore, thermal sensitivity would be weakly phylogenetically heritable. Thermodynamic constraints would prevent large deviations from the central tendency.

## 3. Thermal sensitivity evolves in other ways

This is an "umbrella" hypothesis that encompasses multiple subhypotheses that do not invoke the UTD assumption. For example, a key central tendency (thermodynamic constraint) may still exist, but its influence would be very weak, allowing for a wide exploration of the parameter space away from it. In this case, changes in thermal sensitivity could be the outcome of adaptation to different thermal environments. Another subhypothesis is that clades differ systematically in the rate at which thermal sensitivity evolves, due to the occasional emergence of evolutionary innovations. Thus, clades with high evolutionary rates would be able to better explore the parameter space of thermal sensitivity (i.e., through large changes in $E$ and $W_{op}$ values), compared to low-rate clades in which thermal sensitivity would evolve more gradually. A third subhypothesis is that evolution may favour individuals (and metabolic variants) that are relatively insensitive to temperature fluctuations. In that case, the central tendency of $E$ would not be stationary but moving towards lower values with evolutionary time. It is worth clarifying that these 3 subhypotheses are not necessarily mutually exclusive.

# Results

## Dataset sources

We combined 2 preexisting datasets of $r_{max}$ TPCs, spanning 380 phytoplankton species (a polyphyletic group that includes prokaryotic Cyanobacteria and eukaryotic phyla such as Dinophyta) [35] and 272 prokaryote species (bacteria and archaea) [32]. In addition, we also collected 2 TPC datasets of traits that underlie $r_{max}$: net photosynthesis and respiration rates of algae and aquatic and terrestrial plants (221 and 201 species, respectively) [30]. We used these 2 smaller datasets to understand whether the evolutionary patterns of thermal sensitivity differ between (i) higher-order traits and (ii) traits that are more tightly linked to organismal physiology. Trait values were typically measured under nutrient-, light-, and $CO_2$-saturated conditions (where applicable), after acclimation to each experimental temperature.

To investigate the evolution of measures of thermal sensitivity across species, we reconstructed the phylogeny of as many species in the 4 datasets as possible, from publicly available nucleotide sequences of (i) the small subunit rRNA gene from all species groups and the (ii) cbbL/rbcL gene from photosynthetic prokaryotes, algae, and plants (see the Methods section). We managed to obtain small subunit rRNA gene sequences from 537 species and cbbL/rbcL sequences from 208 of them (Tables D and E in S1 Appendix).

TPC parameters were quantified for each species/strain present in the phylogeny using the Sharpe-Schoolfield model (see Fig 1 and the Methods section). The resulting estimates of $E$ (the slope of the rise of the TPC) and $W_{op}$ (the operational niche width of the TPC) were found to be right-skewed (Fig B in S1 Appendix) as has been shown previously [18, 21]. Furthermore, we did not detect a disproportionately high density of thermal sensitivity values near the lower boundary of $E$ (0 eV), as we would expect if all variation was due to strong measurement error around a true value of, e.g., 0.65 eV. Thus, these results are not consistent with the hypothesis of a nearly invariant thermal sensitivity (hypothesis 1).

## Phylogenetic comparative analyses

We next investigated the evolutionary patterns of thermal sensitivity. Given that the main focus of this study was to investigate how the thermal sensitivity of $r_{max}$ (a direct measure of fitness) evolves, most of the following comparative analyses were performed on our 2 large TPC datasets ($r_{max}$ of phytoplankton and prokaryotes). Besides this, the sample sizes of the 2 smaller datasets would be inadequate for obtaining robust results for many of our analyses. If an analysis makes use of all 4 datasets, this is explicitly stated.

An issue that is worth mentioning is the overlap between the datasets of phytoplankton and prokaryotic TPCs, given that both of them include Cyanobacteria. To address this, we kept Cyanobacteria as part of the phytoplankton dataset (due to their functional similarity) and did not include them in analyses of prokaryotes. We also examined whether our results were mainly driven by the long evolutionary distance between Cyanobacteria and eukaryotic phytoplankton by repeating all phytoplankton analyses after removing Cyanobacteria (see subsection C.2 in S1 Appendix).

**Estimation of phylogenetic heritability.**   As TPC parameters capture different features of the shape of the same curve, it is likely that some of them may covary [35]. To account for this in the estimation of phylogenetic heritability, we fitted a multiresponse phylogenetic regression model using the MCMCglmm R package (version 2.26) [36] in which all TPC parameters formed a combined response. To compare the phylogenetic heritabilities of TPC parameters between planktonic photosynthetic autotrophs and other microbes (autotrophs and heterotrophs), we fitted the model separately to our 2 large TPC datasets: $r_{max}$ of phytoplankton and

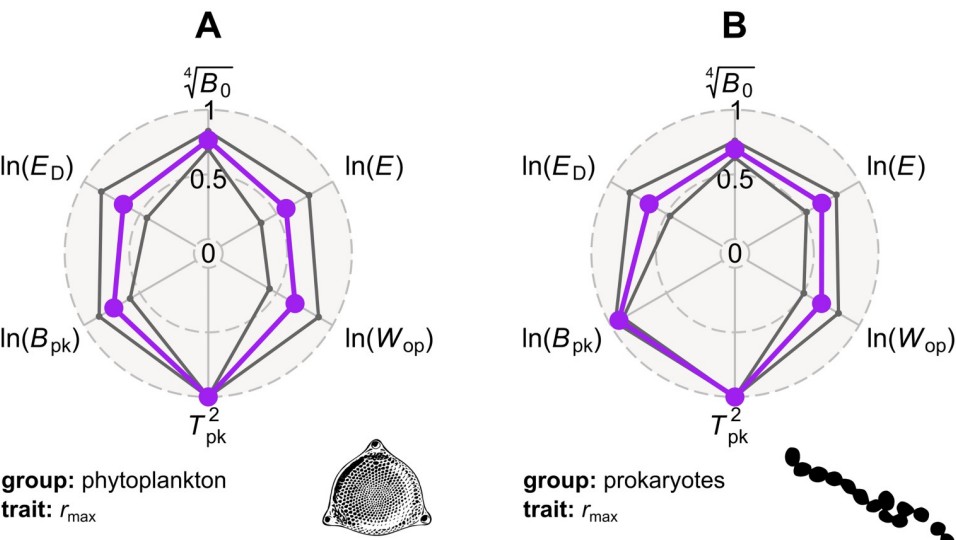

**Fig 2. Moderate to strong phylogenetic heritability can be detected in all TPC parameters, across phytoplankton and prokaryotes.** The 3 circles of each radar chart correspond to phylogenetic heritabilities of 0, 0.5, and 1. Mean phylogenetic heritability estimates—as inferred with MCMCglmm—are shown in purple, whereas the 95% HPD intervals are in dark grey. Note that we transformed all TPC parameters so that their statistical distributions would be approximately Gaussian. The data underlying this figure are available at https://doi.org/10.6084/m9.figshare.12816140. v1. HPD, highest posterior density; TPC, thermal performance curve.

prokaryotes. To satisfy the assumption of models of trait evolution that the change in trait values is normally distributed, we transformed all TPC parameters so that their distributions would be approximately Gaussian (see Fig 2). To ensure that the resulting phylogenetic heritability estimates did not merely reflect the priors that were used in the MCMCglmm analysis, we also estimated phylogenetic heritabilities using the R package Rphylopars (version 0.2.12) [37] and BayesTraits (version 3.0.2) [38]. The main difference between these 2 and MCMCglmm is that Rphylopars and BayesTraits cannot account for the covariance among TPC parameters.

The MCMCglmm analysis revealed the presence of non-negligible phylogenetic heritability in measures of thermal sensitivity ($E$ and $W_{op}$), as well as all other TPC parameters, across phytoplankton (including or excluding Cyanobacteria) and prokaryotes (Fig 2 and Fig J in S1 Appendix). In particular, the phylogenetic heritability estimates of $\ln(E)$ and $\ln(W_{op})$ were statistically different from both 0 and 1, indicating that the 2 TPC parameters evolve across the phylogeny but not in a purely random (Brownian motion) manner. It is worth stressing that even the lower bounds of the 95% highest posterior density (HPD) intervals of $\ln(E)$ and $\ln(W_{op})$ were far greater than 0, allowing us to completely rule out the possibility that all variation in thermal sensitivity is due to measurement error. In general, TPC parameters exhibit a similar phylogenetic heritability between the 2 species groups. The only major exception is $\ln(B_{pk})$, which is considerably more heritable among prokaryotes than among phytoplankton. This difference in phylogenetic heritability most likely reflects the strength of the positive correlation between $B_{pk}$ and $T_{pk}$ (a "hotter is better" pattern) in the 2 groups. More precisely, $T_{pk}$, which has a phylogenetic heritability of $\approx 1$, is more strongly correlated with $B_{pk}$ among prokaryotes [32] than among phytoplankton [35], possibly due to the differences in their cellular physiology. For example, phytoplankton growth rate depends on the interplay among the processes of photosynthesis, respiration, and cell maintenance, whose thermal sensitivities can strongly differ [30]. Qualitatively similar results were obtained from the estimation of

phylogenetic heritabilities with Rphylopars and BayesTraits (S1 Appendix, Fig F). Overall, these results serve as further evidence that hypothesis 1 (that thermal sensitivity does not vary across species) can clearly be rejected.

**Partitioning of thermal sensitivity across the phylogeny.**   To understand why thermal sensitivity has a low to intermediate phylogenetic heritability, we examined how clades throughout the phylogeny explore the parameter space (of $E$ and $W_{op}$) using a disparity-through-time analysis [39, 40]. At each branching point of the phylogeny, mean subclade disparity is calculated as the average squared Euclidean distance among trait values within the subclades, divided by the disparity of trait values across the entire tree. Mean subclade disparity values close to 0 indicate that the mean of the trait variances within subclades is much lower than the variance of trait values across the entire phylogeny. When the opposite occurs, the mean subclade disparity will be close to 1 or even higher. The resulting disparity line is then compared to the null expectation, i.e., an envelope of disparities obtained from simulations of Brownian motion on the same tree. Through the comparison of the observed trait disparity with the null expectation, it is possible to identify the periods of evolutionary time during which mean subclade disparity is higher or lower than expected under Brownian motion. Higher than expected subclade disparity indicates that clades converge in trait space, whereas lower than expected subclade disparity indicates that clades occupy distinct areas of parameter space. The latter pattern is consistent with an adaptive radiation, in which an initial period of rapid trait evolution is typically followed by a deceleration of the evolutionary rate as ecological niches become filled [41, 42]. Frequent episodes of higher than expected subclade disparity (evolutionary convergence) in thermal sensitivity or segregation of major clades in the parameter space would be consistent with hypothesis 3.

The mean subclade disparity of thermal sensitivity measures was considerably higher than expected near the present, highlighting an increasing overlap in the parameter space of thermal sensitivity among distinct clades (Fig 3 and Fig K in S1 Appendix). This pattern of increasing clade-wide convergence in thermal sensitivity is also apparent when comparing the thermal sensitivity distributions of different phyla (Fig 4 and Fig C in S1 Appendix). For example, the distributions of $E$ and $W_{op}$ of Proteobacteria and Bacillariophyta have similar shapes and central tendencies, despite the long evolutionary distance that separates the 2 phyla. This high convergence in thermal sensitivity space by diverse lineages suggests that variation in the 2 TPC parameters is mainly driven by adaptation to local environmental conditions, irrespective of species' evolutionary history. In other words, it is likely that particular thermal strategies (e.g., having low thermal sensitivity) may yield significant fitness gains in certain environments (e.g., those with strong temperature fluctuations that occur predominantly across—rather than within—generations [24, 25]), leading to convergent evolution of thermal sensitivity. It is worth noting that these disparity patterns are not an artefact of a potentially inaccurate tree topology, as higher than expected subclade disparity occurs mainly near the present, where tree nodes have generally high statistical support (S1 Appendix, Fig A).

**Mapping the evolutionary rate on the phylogeny.**   We next investigated whether clades systematically differ in their evolutionary rate for thermal sensitivity (part of hypothesis 3). To this end, we examined the variation in the evolutionary rate of thermal sensitivity measures across the phylogeny by fitting 3 extensions of the Brownian motion model: the free model [43], the stable model [44], and the Lévy model [45]. Under the free model, the trait takes a random walk in the parameter space (Brownian motion) but with an evolutionary rate that varies across branches. The stable model can be seen as a generalisation of the free model, as the evolutionary change in trait values is sampled from a heavy-tailed stable distribution, of which the Gaussian distribution (assumed under Brownian motion) is a special case. Thus, the

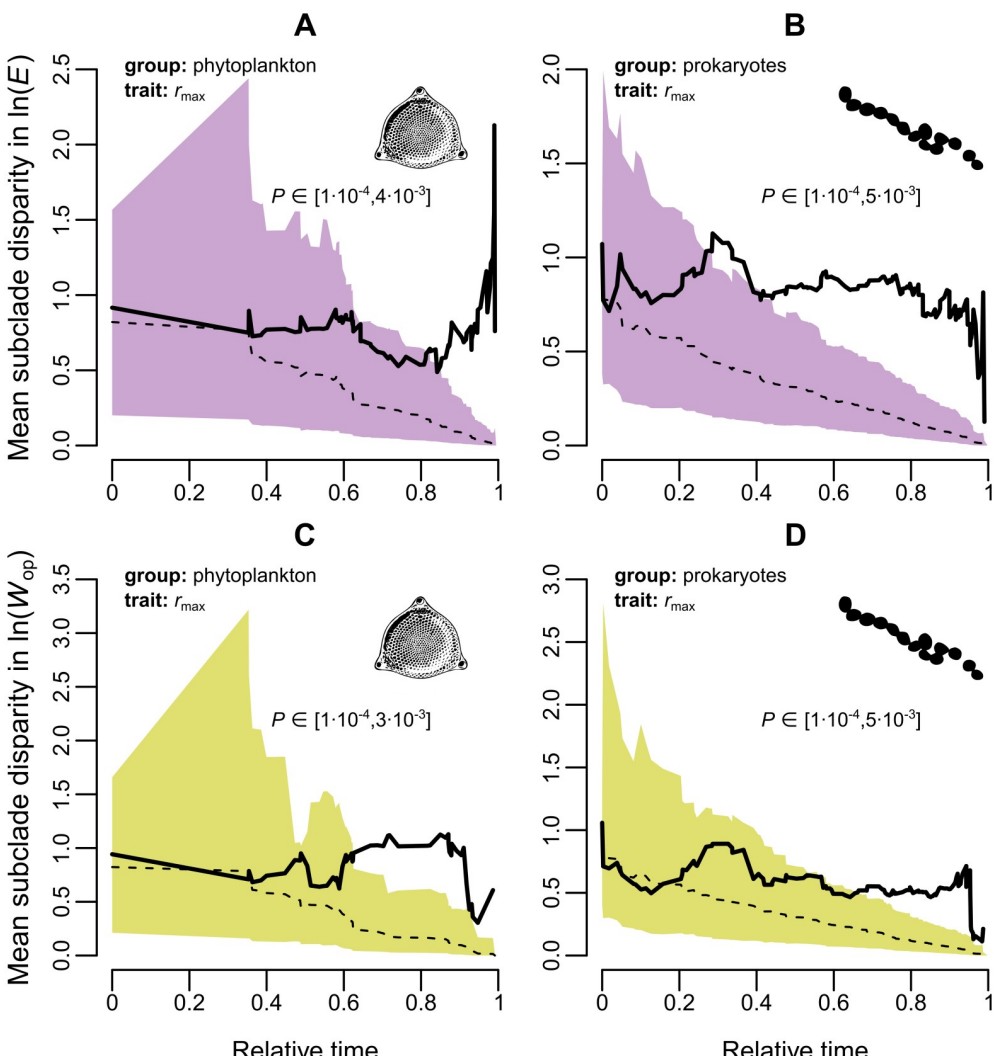

**Fig 3. Change in mean subclade disparity in thermal sensitivity through time.** Shaded regions represent the 95% confidence interval of the resulting trait disparity from 10,000 simulations of random Brownian evolution on each respective subtree (subset of the entire phylogeny). The dashed line stands for the median disparity across simulations, whereas the solid line is the observed trait disparity. The latter is plotted from the root of the tree ($t = 0$) until the most recent internal node. The reported $P$ values were obtained from the rank envelope test [40], whose null hypothesis is that the trait follows a random walk in the parameter space. Note that instead of a single value, a range of $P$ values is produced for each panel, due to the existence of ties. In general, species from evolutionarily remote clades tend to increasingly overlap in thermal sensitivity space (mean subclade disparity exceeds that expected under Brownian motion) with time. The raw data underlying this figure are available at https://doi.org/10.6084/m9.figshare.12816140. v1.

stable model should provide a more accurate representation of evolutionary rate variation, as it is better able to accommodate jumps in parameter space towards rare and extreme trait values. Finally, the Lévy model represents evolution under Brownian motion combined with occasional episodes of rapid trait change.

The results were robust to the choice of model used for inferring evolutionary rates (Fig 5, Figs G and H in S1 Appendix). Rate shifts tend to occur sporadically throughout the phylogeny and especially in late-branching lineages, without being limited to particular clades. This pattern suggests that there is little systematic variation in the evolutionary rate of thermal

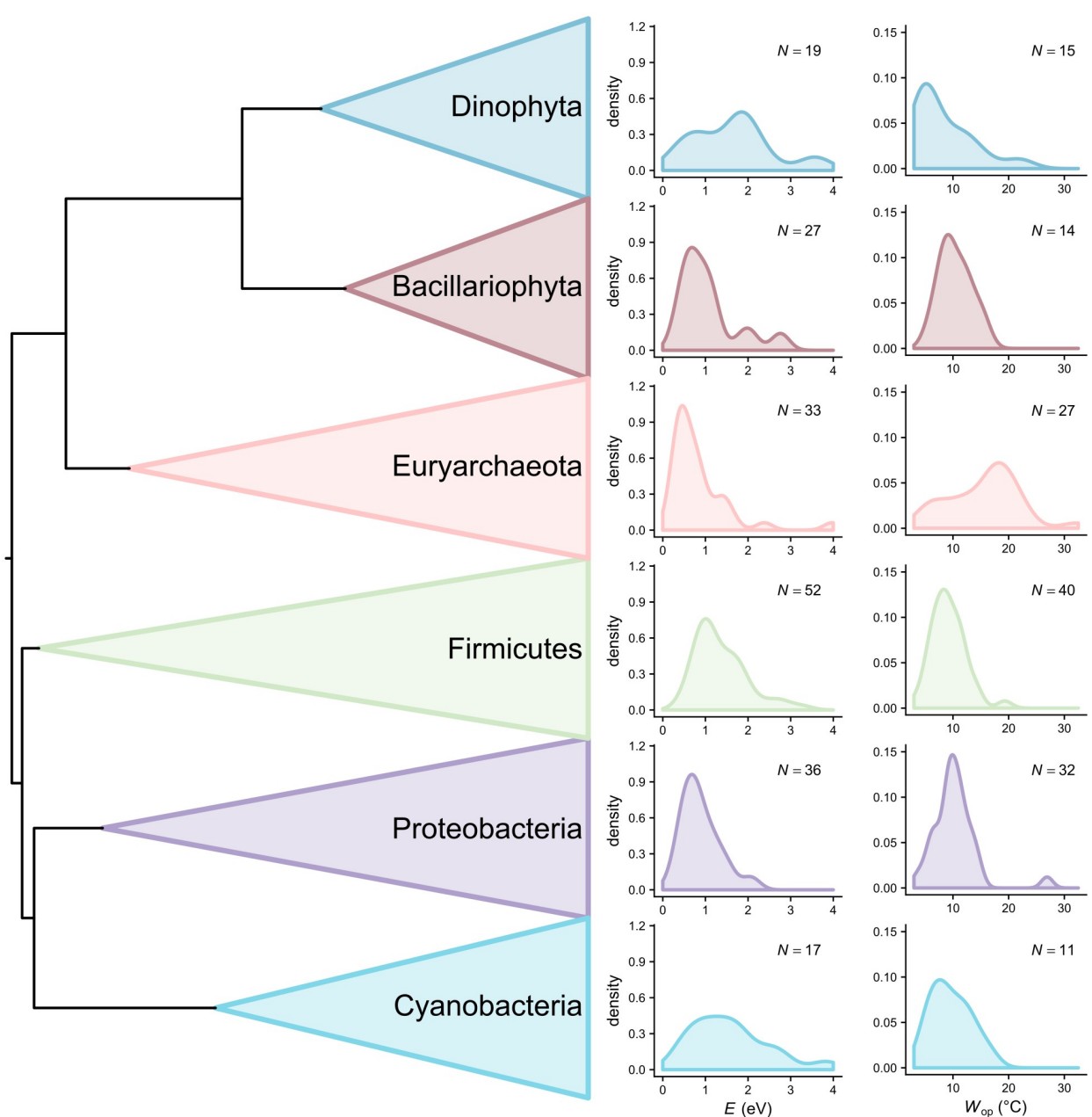

**Fig 4. Distributions of thermal sensitivity estimates of $r_{max}$ for the largest (most species-rich) phyla of this study.** In general, more variation can be observed within than among phyla. The data underlying this figure are available at https://doi.org/10.6084/m9.figshare.12816140.v1.

sensitivity among clades, with sudden bursts of trait evolution arising in parallel across evolutionarily remote lineages.

**Visualization of trait evolution as a function of time, and test for directional selection.** To further describe the evolution of thermal sensitivity, we visualized the $E$ and $W_{op}$ values from the root of each subtree until the present day, across all 4 TPC datasets, using the phytools R package (version 0.6–60) [46]. Ancestral states—and the uncertainty around them—were obtained from fits of the stable model of trait evolution, as described in the

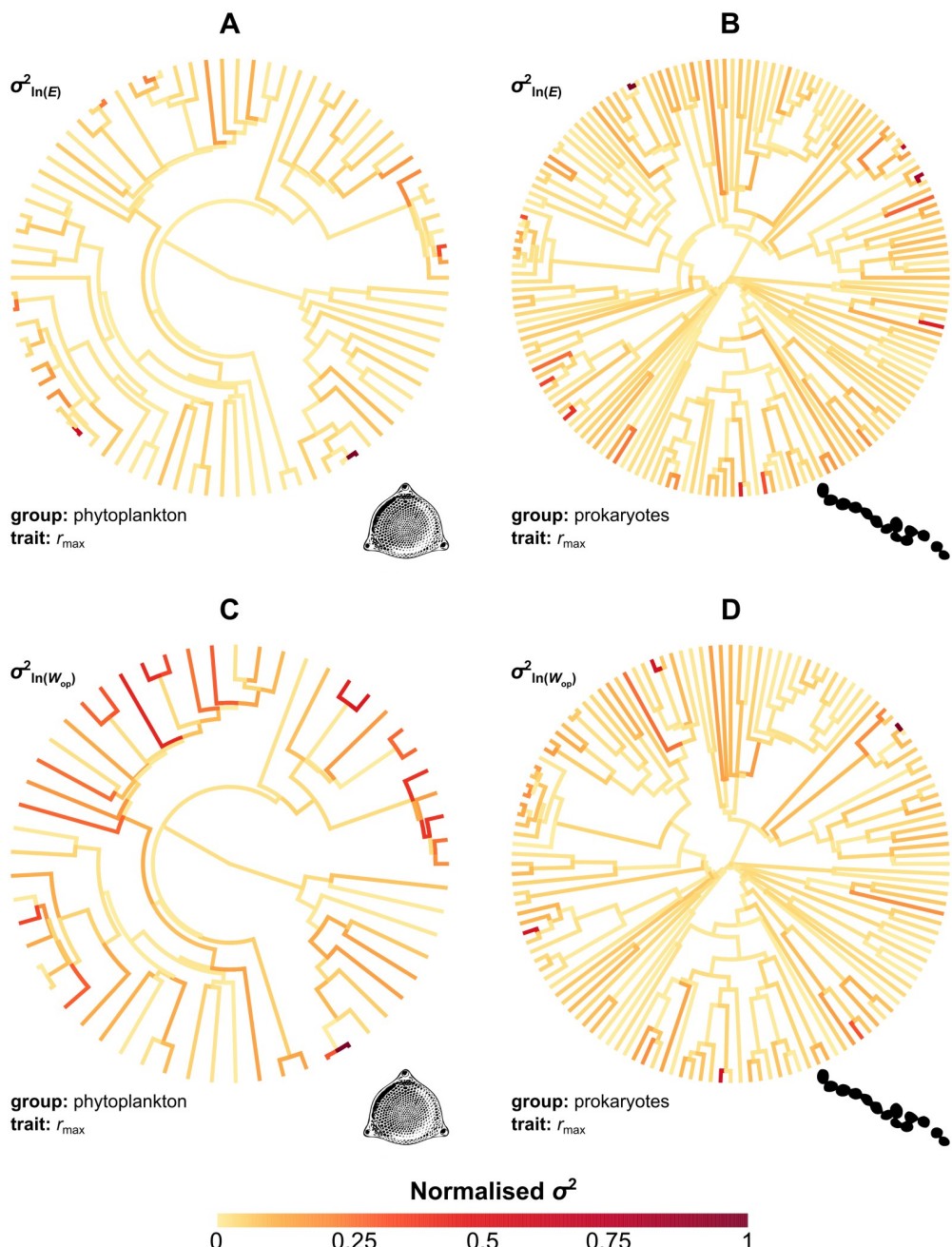

**Fig 5. Variation in the evolutionary rate of thermal sensitivity across the phylogeny.** Rates were estimated by fitting the stable model of trait evolution to each dataset and were then normalised between 0 and 1. Most branches exhibit relatively low rates of evolution (orange), whereas the highest rates (red and brown) are generally observed in late-branching lineages across different clades. The raw data underlying this figure are available at https://doi.org/10.6084/m9.figshare.12816140.v1.

previous subsection. The visualization allowed us to test hypothesis 2, i.e., that thermal sensitivity evolves around a central tendency of 0.65 eV (or 0.32 eV), with large deviations from this value reverting quickly back to it. To this end, and to also test the hypothesis of directional selection towards lower thermal sensitivity (part of hypothesis 3), we used the following

model:

$$\ln(E) \sim \ln(\hat{\theta}) + \text{slope} \cdot t. \tag{2}$$

$\ln(E)$ values (those from extant species and ancestral states inferred with the stable model) were regressed against a central value ($\ln(\hat{\theta})$) and a slope that captures a putative linear trend towards lower/higher values with relative time, $t$. The same model was also fitted to $\ln(W_{op})$. The regressions were performed with MCMCglmm and were corrected for phylogeny as this resulted in lower deviance information criterion (DIC) [47] values than those obtained from non-phylogenetic variants of the models. More precisely, we executed 2 MCMCglmm chains per regression for a million generations, sampling every thousand generations after the first hundred thousand.

This analysis (Fig 6) did not provide support for the hypothesis of strongly constrained adaptive evolution around a single key central value (hypothesis 2). Instead, lineages explore large parts of the parameter space, often moving rapidly towards the upper and lower bounds (i.e., 0 and 4 eV), without reverting back to the presumed central tendency (e.g., see the clade denoted by the arrow in Fig 6D). The estimated central values for $E$ of the two $r_{max}$ datasets were much higher than the MTE expectation, and, in the case of prokaryotes (Fig 6B), the 95% HPD interval did not include 0.65. Similarly, the inferred central values for $E$ of net photosynthesis rate and respiration rate (0.52 eV and 2.06 eV, respectively; Fig La,b in S1 Appendix) were both higher than 0.32 and 0.65 eV. The slope parameter that would capture the effects of directional selection in thermal sensitivity (part of hypothesis 3) was not statistically different from 0 for any dataset.

## Latitudinally structured variation in thermal sensitivity

All our analyses so far converge on one conclusion: that the evolution of the thermal sensitivities of fitness-related traits can be rapid and largely independent of the evolutionary history of each lineage. This suggests that certain environments may select for particular values of thermal sensitivity. To identify environmental adaptation in thermal sensitivity, we tested for latitudinal variation in it using the combination of all 4 TPC datasets. Specifically, the increase in temperature fluctuations from low to intermediate absolute latitudes is expected to increasingly select for thermal generalists (lower $E$ and higher $W_{op}$ values) [3, 48, 49]. At high latitudes, however, temperature fluctuations may further increase or progressively decrease, depending on environment type (marine versus terrestrial) and differences between the 2 hemispheres [3, 48, 49]. In any case, the overwhelming majority of our thermal sensitivity estimates belonged to species/strains from low and intermediate latitudes (S1 Appendix, Fig M), enabling us to investigate the hypothesized gradual transition towards lower thermal sensitivity from the equator to intermediate latitudes.

Latitude indeed explained a significant amount of variation in $E$ (which declined as expected) but not in $W_{op}$ (Fig 7 and Fig N in S1 Appendix, Tables A and B in S1 Appendix). The $E$ estimates of $r_{max}$, net photosynthesis rate, and respiration rate differed statistically in their intercepts but not in their slopes against latitude, although the latter could be an artefact of the small sample size. This result suggests that latitude could influence the $E$ values of not only $r_{max}$ but also other traits across various species groups. Dividing latitude into 3 bins (i.e., low, intermediate, and high absolute latitudes) and comparing their $E$ distributions yielded similar conclusions (S1 Appendix, Fig O, Table C).

We also tested for a possible latitudinal clade age bias, which could arise if certain clades originated in particular latitudes and only much later expanded to other areas [50, 51]. For this, we performed a Mantel test [52] to estimate the correlation between phylogenetic distance

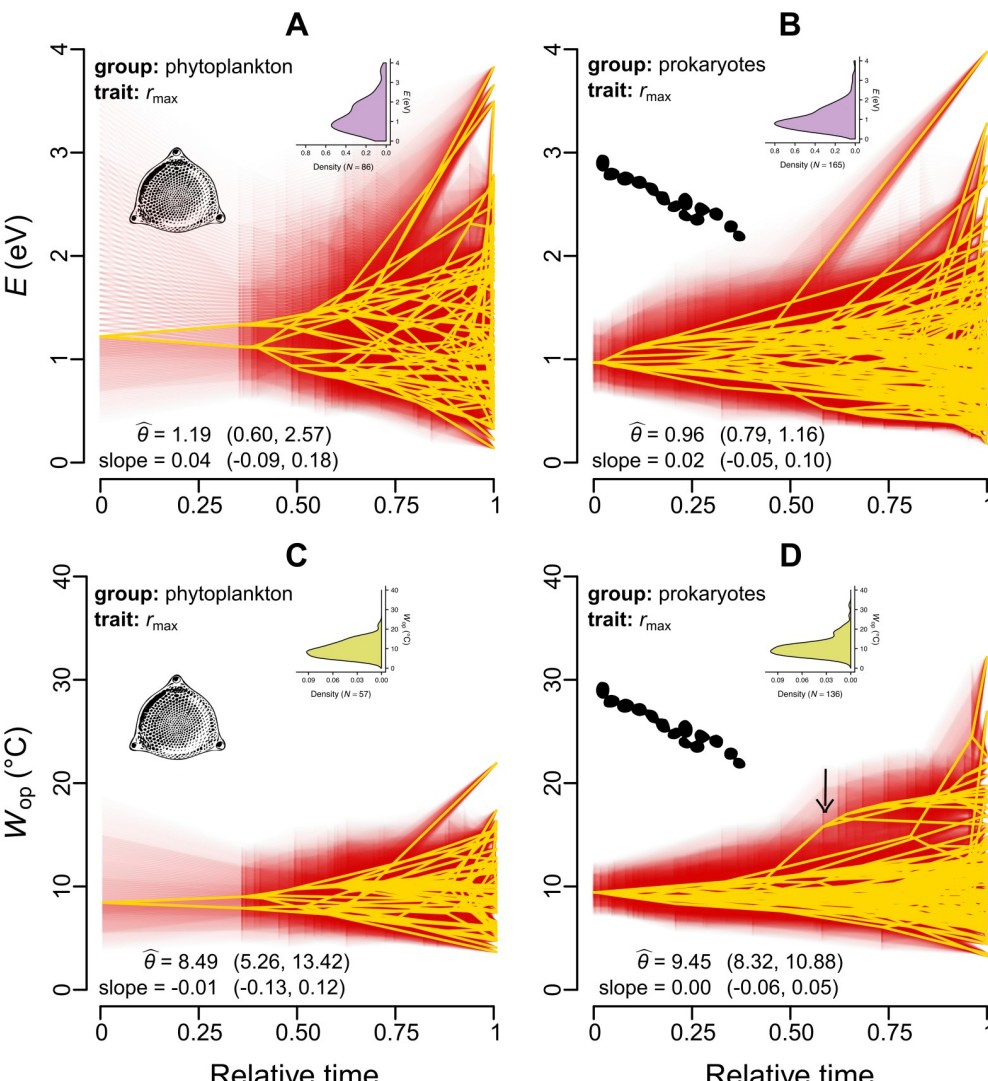

**Fig 6. Projection of the phylogeny into thermal sensitivity versus time space.** The values of ancestral nodes were estimated from fits of the stable model. Yellow lines represent the median estimates, whereas the 95% credible intervals are shown in red. $\hat{\theta}$ is the estimated central tendency for each panel, whereas the existence of a linear trend towards lower/higher values is captured by the reported slope. Parentheses stand for the 95% HPD intervals for $\hat{\theta}$ and the slope. All estimates were obtained for $\ln(E)$ and $\ln(W_{op})$, but the parameters are shown here in linear scale. The inset figures show the density distributions of $E$ and $W_{op}$ values of extant species in the dataset. The arrow in panel D shows an example of a whole clade shifting towards high $W_{op}$ values, without being attracted back to $\hat{\theta}$. The raw data underlying this figure are available at https://doi.org/10.6084/m9.figshare.12816140.v1. HPD, highest posterior density.

and latitudinal distance for the 2 largest groups of our study (phytoplankton and prokaryotes). No such bias was detected for phytoplankton ($r = 0.04$, $P = 0.114$), whereas, for prokaryotes, the correlation was statistically supported but very weak ($r = 0.11$, $P = 0.002$). This result indicates that neither species group is characterised by very strong dispersal limitation throughout its evolutionary history.

## Discussion

In this study, we have performed a thorough analysis of the evolution of the thermal sensitivities of $r_{max}$ in phytoplankton and prokaryotes and its 2 key underlying physiological traits in

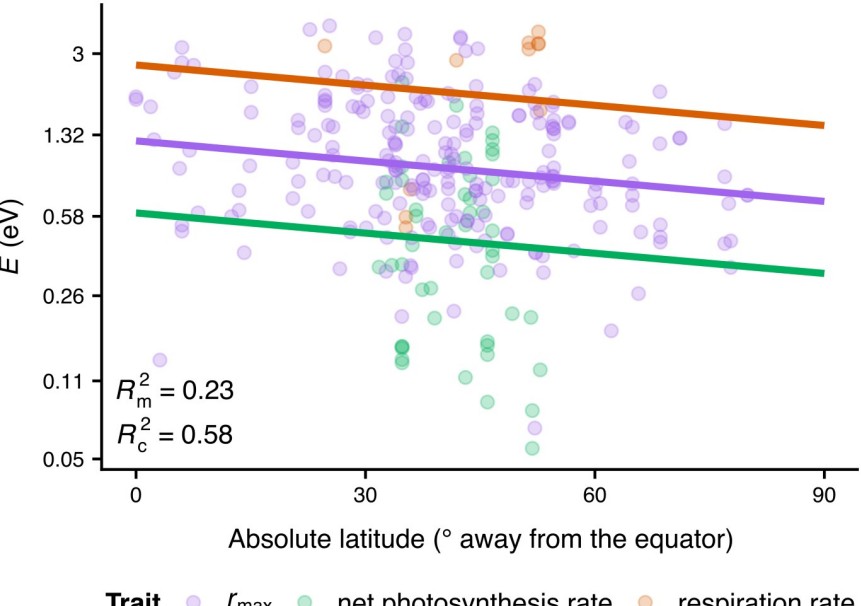

**Fig 7. *E* values weakly decrease with absolute latitude.** 23% of the variance is explained by latitude and trait identity, which increases to 58% if species identity is added as a random effect on the intercept. Note that values on the vertical axis increase exponentially. The data underlying this figure are available at https://doi.org/10.6084/m9.figshare.12816140.v1.

plants (net photosynthesis rate and respiration rate). To achieve this, we formulated and tested 3 alternative hypotheses that represent different views expressed in the literature regarding the impact of thermodynamic constraints on the evolution of thermal sensitivity of fitness-related traits.

The first hypothesis was that the activity of a single key rate-limiting enzyme of respiration or photosynthesis directly determines the performance of physiological traits [9, 13] and emergent proxies for fitness (such as $r_{max}$) [53, 54] (the UTD assumption). As a result, thermal sensitivity should be strictly constant across traits, species, and environments. This hypothesis was first introduced in early papers that described the MTE [8–10]. In contrast to the UTD expectation, we detected substantial variation in thermal sensitivity, within and across traits and species groups (Fig B in S1 Appendix). Furthermore, the distribution of *E* (slope of the rising part of the TPC) values did not exhibit an inflated density near its lower boundary (around 0 eV), as we would expect if all variation in thermal sensitivity was due to measurement error. The rejection of hypothesis 1 was additionally supported by our finding that thermal sensitivity is phylogenetically heritable across phytoplankton and prokaryotes (Fig 2).

Our second hypothesis was that thermal sensitivity evolves across species but remains close to a key value imposed by strong (but not insurmountable) thermodynamic constraints. We tested this hypothesis using a series of phylogenetic comparative analyses which revealed that the evolution of thermal sensitivity is characterised by an increasing overlap in parameter space by evolutionarily remote lineages (Figs 3 and 4) due to bursts of rapid evolution (Fig 5). Additionally, visualisation of thermal sensitivity evolution through time (Fig 6 and Fig L in S1 Appendix) showed that thermal sensitivity can rapidly move away from its presumed central value without being strongly attracted back to it (e.g., see the arrow in Fig 6D). In conclusion, these results lead us to reject hypothesis 2, i.e., that thermal sensitivity evolves under very strong thermodynamic constraints.

Our final hypothesis was that thermal sensitivity evolves in an adaptive manner and that, even if a central tendency exists, its influence on thermal sensitivity evolution is weak. This hypothesis was supported by the results of all phylogenetic comparative analyses and by our detection of a systematic relationship between $E$ and latitude. The latter is likely driven by the increase in temperature fluctuations from the equator to intermediate latitudes and agrees with the expectation that thermally variable environments should select for phenotypes with low thermal sensitivity, and vice versa [3, 21, 24, 25]. That a similar latitudinal effect could not be detected on $W_{op}$ (the operational niche width of the TPC) is possibly because of the much smaller sample size available for this, combined with the fact that $W_{op}$ is nonlinearly related to $E$ (S1 Appendix, Fig D). More precisely, in the Sharpe-Schoolfield model, $W_{op}$ will necessarily decrease as $E$ increases, provided that changes in $E$ are not strongly associated with changes in other TPC parameters (e.g., $T_{pk}$ or $E_D$; Fig E in S1 Appendix). Indeed, a previous study showed that $E$ correlates systematically only with $W_{op}$ [35]. In any case, $E$ is arguably a more meaningful measure of thermal sensitivity than $W_{op}$ because the latter assumes that species mainly experience temperatures close to $T_{pk}$, while $E$ captures the entire rise of the TPC. Besides temperature fluctuations, a decrease in $E$ with absolute latitude could also be explained by the metabolic cold adaptation hypothesis [55–58]. According to it, cold-adapted species should evolve lower thermal sensitivities (as well as higher $B_0$ values; see Fig 1) to maintain sufficient trait performance at very low temperatures. As our datasets do not possess the necessary resolution (especially at high latitudes; Figs M and O in S1 Appendix) for differentiating between these 2 alternative (and non-mutually exclusive) processes, this question remains to be addressed by future research.

Overall, a set of novel mechanistic explanations of TPC evolution emerge from our comparison of phylogenetic heritabilities of TPC parameters (Fig 2 and Fig F in S1 Appendix). Contrary to $E$ and $W_{op}$, which have low to intermediate phylogenetic heritabilities, $T_{pk}$ is almost perfectly phylogenetically heritable and evolves relatively gradually (i.e., without large jumps in parameter space; see Fig I in S1 Appendix). Thus, we expect TPCs to adapt to different thermal environments through both gradual changes in $T_{pk}$ and discontinuous changes in $E$. Gradual changes in $T_{pk}$ may be achieved through evolutionary shifts in the melting temperature of enzymes, i.e., the temperature at which 50% of the enzyme population is deactivated [59, 60]. In contrast, changes in thermal sensitivity may be the outcome of (i) evolution of enzymes with different heat capacities [60–62], (ii) changes in the plasticity of cellular membranes [3, 63], or even (iii) restructuring of the underlying metabolic network [64].

Fundamental differences in the selection mechanisms underlying the evolution of $T_{pk}$ and $E$ may also explain the difference in evolutionary patterns between them. Specifically, both the mean environmental temperature (to which $T_{pk}$ responds [7, 35]) and the temperature fluctuations (to which $E$ responds [3, 21, 24–26, 35]) vary systematically from the equator to intermediate latitudes. We hypothesize that a species adapted to low temperatures is unlikely to adapt to a high-temperature environment rapidly enough (i.e., through a large increase in $T_{pk}$) as it is pushed to its thermal tolerance limits [65, 66]. In contrast, a species adapted to a fluctuating thermal environment (i.e., with a low $E$ value) should be able to survive in more thermally stable conditions without much cost, becoming a thermal specialist (i.e., with a high $E$ value) relatively rapidly, resulting in the observed jumps in trait space when mapped on the phylogeny (Figs 3 and 6, Figs K and L in S1 Appendix).

It is worth stressing, however, that not all types of thermal fluctuations are expected to impose selection for thermal generalists. In particular, thermal generalist variants of a given species are expected to be favoured when temperature fluctuations primarily occur across generations [24, 25, 67]. In contrast, moderate to strong thermal variation within generations would lead to selection for thermal specialists, even when intergenerational fluctuations are

also present. For the microbial groups of the present study, an estimate of the minimum generation time can be calculated as the inverse of the $B_{pk}$ of $r_{max}$. Across our datasets of phytoplankton and prokaryotes, the minimum generation time ranges from a few minutes to 3.5 months, with phylogenetically corrected medians of $\approx$ 40.5 hours for phytoplankton and $\approx$ 3.5 hours for prokaryotes (S1 Appendix, Fig P). Given this and because the magnitudes of annual and intra-annual (e.g., monthly) thermal fluctuations increase from the equator to intermediate latitudes [3, 48, 68], most microbes from intermediate latitudes are expected to generally experience substantial intergenerational thermal fluctuations and to a much lesser extent intragenerational fluctuations. This is indeed consistent with the observed weak decline in $E$ at intermediate latitudes compared to the equator (Fig 7 and Fig O in S1 Appendix). Nevertheless, latitude, trait identity, and species identity account for only 58% of the variance in $E$, indicating that adaptive shifts in $E$ may also be driven by other factors such as biotic interactions [18, 69, 70]. A systematic identification of drivers of thermal sensitivity as well as the magnitude of their respective influence could be the focus of future studies.

For the thermal sensitivity of $r_{max}$ in particular, the observed patterns of discontinuous evolution likely reflect the evolution of TPCs of underlying physiological traits on which it depends. For example, in populations of photosynthetic cells, shifts in the thermal sensitivity of any or all of photosynthesis rate, respiration rate, and carbon allocation efficiency can induce large changes in the $E$ of $r_{max}$ [30]. Indeed, we observed large adaptive shifts in thermal sensitivity even for fundamental physiological traits such as respiration rate (S1 Appendix, Fig Lb,d), contrary to the MTE expectation of strong evolutionary conservatism [8–10]. This result is in agreement with a previous study that had identified significant adaptive variation in the TPC of the specific activity of Rubisco carboxylase [31]. It remains to be seen whether a similar lack of evolutionary conservation can be detected in key enzymes of non-photosynthetic organisms. Further research is clearly also needed on how the thermal sensitivities of different traits underlying fitness interact, and the extent to which these interactions can be modified through adaptation.

Besides biological-driven variation in thermal sensitivity, "artificial" variation may also be present, hindering the recognition of real patterns. For example, $E$ estimates can be inaccurate if trait measurements in the rise of the TPC are limited, and span too narrow a range of temperatures [12]. To address this issue, we only kept $E$ estimates if at least 4 trait measurements were available at the rise of each TPC. Further variation in thermal sensitivity can be introduced if trait values are measured instantaneously (without allowing sufficient time for acclimation) or under suboptimal conditions (e.g., under nutrient- or light-deficient conditions). Such treatments can lead to systematic biases in the shape of the resulting TPCs, which may strongly differ from TPCs obtained after adequate acclimation and under optimal growth conditions [27, 71–74]. To avoid such biases, the datasets that we used only included TPCs that were experimentally determined after acclimation and under optimal conditions. On the other hand, maintenance of a given strain under a fixed set of experimental conditions for hundreds of generations could also lead to adaptive changes in TPC shape, due to the emergence of novel genetic mutations, as has been previously shown [26, 27]. While the strains in our dataset were not grown over such long time periods, future studies could employ experimental evolution to measure the rate of thermal sensitivity evolution over much shorter timescales than the ones in our study.

Put together, all these results yield a compelling mechanistic explanation of how evolution shapes the distribution of $E$ and emphasize the need to consider the ecological and evolutionary underpinnings as well as implications of variation in $E$, as has been pointed out in a spate of recent studies [12, 18, 20, 21, 30]. In particular, our study helps explain the reason for the right skewness in the $E$ distributions previously identified across practically all traits and

taxonomic groups [12, 18, 21]. A clear explanation for this pattern has been lacking, partly because MTE posits that $E$ should be thermodynamically constrained and thus almost invariable across species [8–10]. Our study fills this gap in understanding by showing that the distribution of $E$ is the outcome of frequent convergent evolution, driven by the adaptation of species from different clades to similar environmental conditions. In other words, as species encounter new environments through active or passive dispersal [75–77], they face selection for particular values of thermal sensitivity, which results in (often large) shifts in $E$. This process explains both the low variation in $E$ among some species groups (Fig 4) and the shape of its distribution. More precisely, the high degree of right skewness probably reflects the fact that most environments select for thermal generalists, with high $E$ values being less frequently advantageous. Our findings have implications for ecophysiological models, which may benefit from accounting for variation in thermal sensitivity among species or individuals. This could both yield an improved fit to empirical datasets [78] and provide a more realistic approximation of the processes being studied. Finally, the existence of adaptive variation in thermal sensitivity is likely to partly drive ecological patterns at higher scales (e.g., the response of an ecosystem to warming). How differences in thermal sensitivity among species influence ecosystem function is largely unaddressed [32, 78] but highly important for accurately predicting the impacts of climate change on diverse ecosystems.

## Methods

### Phylogeny reconstruction and relative time calibration

We performed sequence alignment using MAFFT (version 7.123b) [79] and its L-INS-i algorithm, and we ran Noisy (version 1.5.12) [80] with the default options to identify and remove phylogenetically uninformative homoplastic sites. For a more robust phylogenetic reconstruction, we used the results of previous phylogenetic studies by extracting the Open Tree of Life [81] topology for the species in our dataset using the rotl R package [82]. We manually examined the topology to eliminate any obvious errors. In total, 497 species were present in the tree, whereas many nodes were polytomic. To add missing species and resolve polytomies, we inferred 1,500 trees with RAxML (version 8.2.9) [83] from our concatenated sequence alignment, using the Open Tree of Life topology as a backbone constraint and the General Time-Reversible model [84] with $\Gamma$-distributed rate variation among sites [85]. This model was fitted separately to each gene partition (i.e., one partition for the alignment of the small subunit rRNA gene sequences and one partition for the alignment of cbbL/rbcL gene sequences). Out of the 1,500 resulting tree topologies, we selected the tree with the highest log-likelihood and performed bootstrapping (using the extended majority-rule criterion) [86] to evaluate the statistical support for each node.

Finally, we calibrated the resulting RAxML tree to units of relative time by running DPPDiv [87] on the alignment of the small subunit rRNA gene sequences using the uncorrelated $\Gamma$-distributed rates model [88] (S1 Appendix, Fig A). For this, we used the alignment of small subunit rRNA gene sequences only, as DPPDiv can only be run on a single gene partition. We executed 2 DPPDiv runs for 9.5 million generations, sampling from the posterior distribution every 100 generations. After discarding the first 25% of samples as burn-in, we ensured that the 2 runs had converged on statistically indistinguishable posterior distributions by examining the effective sample size and the potential scale reduction factor [89, 90] for all model parameters. More precisely, we verified that all parameters had an effective sample size above 200 and a potential scale reduction factor value below 1.1. To summarise the posterior distribution of calibrated trees into a single relative chronogram, we kept 4,750 trees per run (one

tree every 1,500 generations) and calculated the median height for each node using the
TreeAnnotator program [91].

## Sharpe-Schoolfield model fitting

To obtain estimates of the parameters of each experimentally determined TPC, we fitted the
following 4-parameter variant of the Sharpe-Schoolfield model (Fig 1) [5, 35]:

$$B(T) = B_0 \cdot \frac{e^{\left[\frac{-E}{k}\left(\frac{1}{T} - \frac{1}{T_{\text{ref}}}\right)\right]}}{1 + \frac{E}{E_D - E} \cdot e^{\left[\frac{E_D}{k}\left(\frac{1}{T_{\text{pk}}} - \frac{1}{T}\right)\right]}} \cdot \tag{3}$$

This model extends the Boltzmann-Arrhenius model (Eq 1) to capture the decline in trait
performance after the TPC reaches its peak ($T_{\text{pk}}$). We followed the same approach for fitting
the Sharpe-Schoolfield model as Kontopoulos and coworkers [35]. Briefly, we set $T_{\text{ref}}$ to 0 ˚C
because, for $B_0$ to be biologically meaningful (see Fig 1), it needs to be normalised at a temper-
ature below the minimum $T_{\text{pk}}$ in the study. Thus, a $T_{\text{ref}}$ value of 0 ˚C allowed us to include
TPCs from species with low $T_{\text{pk}}$ values in the analyses. Also, as certain specific TPC parameter
combinations can mathematically lead to an overestimation of $B_0$ compared to the true value,
$B(T_{\text{ref}})$ [92], we manually recalculated $B(T_{\text{ref}})$ for each TPC after obtaining estimates of the 4
main parameters ($B_0$, $E$, $T_{\text{pk}}$, and $E_D$). For simplicity, these recalculated $B(T_{\text{ref}})$ values are
referred to as $B_0$ throughout the study. Finally, $B_{\text{pk}}$ and $W_{\text{op}}$ were calculated based on the esti-
mates of the 4 main parameters.

After rejecting fits with an $R^2$ below 0.5, there were (i) 312 fits across 118 species from the
phytoplankton $r_{\text{max}}$ dataset, (ii) 289 fits across 189 species from the prokaryote $r_{\text{max}}$ dataset,
(iii) 87 fits across 38 species from the net photosynthesis rates dataset, and (iv) 34 fits across 18
species from the respiration rates dataset. Note that some species were represented by multiple
fits due to the inclusion of experimentally determined TPCs from different strains of the same
species or from different geographical locations. To ensure that all TPC parameters were reli-
ably estimated, we performed further filtering based on the following criteria: (i) $B_0$ and $E$ esti-
mates were rejected if fewer than 4 experimental data points were available below $T_{\text{pk}}$. (ii)
Extremely high $E$ estimates (i.e., above 4 eV) were rejected. (iii) $W_{\text{op}}$ values were retained if at
least 4 data points were available below $T_{\text{pk}}$ and 2 after it. (iv) Two data points below and after
the peak were required for accepting the estimates of $T_{\text{pk}}$ and $B_{\text{pk}}$. (v) $E_D$ estimates were kept if
at least 4 data points were available at temperatures greater than $T_{\text{pk}}$.

## Estimation of phylogenetic heritability for all TPC parameters using MCMCglmm, Rphylopars, and BayesTraits

For MCMCglmm, the methodology that we used was also identical to that of Kontopoulos and
coworkers [35]. In short, we specified a phylogenetic mixed-effects model for each of the 2
large TPC datasets. The models had a combined response with all TPC parameters trans-
formed towards normality. The uncertainty for each estimate was obtained with the delta
method [93] or via bootstrapping (for $\ln(W_{\text{op}})$) and was incorporated into the model. Missing
estimates in the response variables (i.e., when not all parameter estimates could be obtained
for the same TPC) were modelled according to the "Missing At Random" approach [36, 94].
Regarding fixed effects, a separate intercept was specified for each TPC parameter. Species
identity was treated as a random effect on the intercepts and was corrected for phylogeny
through the integration of the inverse of the phylogenetic variance-covariance matrix. For
each dataset, 2 Markov chain Monte Carlo chains were run for 200 million generations, and

estimates of the parameters of the model were sampled every 1,000 generations after the first 20 million generations were discarded as burn-in. Tests to ensure that the chains had converged and that the parameters were adequately sampled were done as previously described.

We also estimated Pagel's $\lambda$ [95] (which is equivalent to phylogenetic heritability [96]) for each TPC parameter using Rphylopars and BayesTraits. For the latter, we executed 2 Markov chain Monte Carlo chains for 10 million generations, kept samples from the posterior every 1,000 generations after the first million, and ensured that sufficient convergence had been reached. Nevertheless, we note that our previous approach is superior because Rphylopars and BayesTraits analyse each TPC parameter separately, and thus covariances among TPC parameters are not taken into account when estimating missing values. Furthermore, these 2 methods cannot accommodate the uncertainty for each TPC parameter estimate.

### Disparity-through-time analyses

We performed disparity-through-time analyses for $\ln(E)$ and $\ln(W_{op})$, using the rank envelope method [40] to generate a confidence envelope from 10,000 simulations of random evolution (Brownian motion). As it is not straightforward to incorporate multiple measurements per species with this method, we selected the $\ln(E)$ or $\ln(W_{op})$ estimate of the Sharpe-Schoolfield model fit with the highest $R^2$ value per species.

### Free, stable, and Lévy model fitting

We fitted the free, stable, and Lévy models of trait evolution to estimates of $\ln(E)$ and $\ln(W_{op})$, using the motmot.2.0 R package (version 1.1.2) [97, 98], the stabletraits software [44], and the levolution software [45], respectively. To obtain each fit of the stable model, we executed 4 independent Markov chain Monte Carlo chains for 30 million generations, recording posterior parameter samples every 100 generations. Samples from the first 7.5 million generations were excluded, whereas the remaining samples were examined to ensure that convergence had been achieved. For fitting the Lévy model, we used the peak-finder algorithm to estimate the value of the model's $\alpha$ parameter. More precisely, we set the starting value of $\alpha$ to $10^{0.5}$, the step size to 0.5, and the number of optimizations to 5, as suggested in levolution's documentation. We also changed the maximum number of iterations (option "-maxIterations") to 2,000 so that the algorithm could sufficiently converge in all cases.

### Investigation of a putative relationship between latitude and $\ln(E)$ and $\ln(W_{op})$

We examined the relationship of thermal sensitivity with latitude by fitting regression models with MCMCglmm to all 4 TPC datasets combined. The response variable was $\ln(E)$ or $\ln(W_{op})$, whereas possible predictor variables were (i) latitude (in radian units and using a cosine transformation, as absolute latitude in degree units, or split in 3 bins of low, intermediate, and high absolute latitude; subsections D.2 and D.3 in S1 Appendix), (ii) the trait from which thermal sensitivity estimates were obtained, and (iii) the interaction between latitude and trait identity. To properly incorporate multiple measurements from the same species (where available), we treated species identity as a random effect on the intercept. We fitted both phylogenetic and non-phylogenetic variants of all candidate models. Two chains per model were run for 5 million generations each, with samples from the posterior being captured every thousand generations. We verified that each pair of chains had sufficiently converged, after discarding samples from the first 500,000 generations. To identify the most appropriate model, we first rejected models that had a nonintercept coefficient with a 95% HPD interval that included 0. We then selected the model with the lowest mean DIC value. To report the proportions of

variance explained by the fixed effects ($\text{Var}_{\text{fixed}}$), by the random effect ($\text{Var}_{\text{random}}$), or left unexplained ($\text{Var}_{\text{resid}}$), we calculated the marginal and conditional coefficients of determination [99]:

$$R_m^2 = \frac{\text{Var}_{\text{fixed}}}{\text{Var}_{\text{fixed}} + \text{Var}_{\text{random}} + \text{Var}_{\text{resid}}}, \qquad (4)$$

$$R_c^2 = \frac{\text{Var}_{\text{fixed}} + \text{Var}_{\text{random}}}{\text{Var}_{\text{fixed}} + \text{Var}_{\text{random}} + \text{Var}_{\text{resid}}}. \qquad (5)$$

### Mantel test between phylogenetic and latitudinal distance matrices

We used the R package ade4 (version 1.7–13) [100] to infer the correlation of phylogenetic distance with latitudinal distance across phytoplankton and prokaryotes using the Mantel test. To generate the $P$ values, we set the number of permutations to 9,999.

## Supporting information

**S1 Appendix. Supplementary material.**
(PDF)

## Acknowledgments

We thank Iain Colin Prentice for providing comments on an early version of the manuscript and Andrew Meade for answering our questions regarding the BayesTraits analysis. We are also grateful to James Rosindell, Jonathan Lloyd, Guy Woodward, and Andrew G. Hirst for valuable discussions, and to the CIPRES Science Gateway [101] for access to computational resources. Species silhouettes were obtained from phylopic.org and are used under the Public Domain license. The images of species in Fig 1 were graciously provided by Eric Erbe, Christopher Pooley, and Rocky Mountain National Park, also under the Public Domain license.

## Author Contributions

**Conceptualization:** Dimitrios—Georgios Kontopoulos, Samraat Pawar.

**Data curation:** Dimitrios—Georgios Kontopoulos.

**Formal analysis:** Dimitrios—Georgios Kontopoulos.

**Funding acquisition:** Samraat Pawar.

**Investigation:** Dimitrios—Georgios Kontopoulos.

**Methodology:** Dimitrios—Georgios Kontopoulos.

**Project administration:** Dimitrios—Georgios Kontopoulos.

**Resources:** Thomas P. Smith.

**Software:** Dimitrios—Georgios Kontopoulos.

**Supervision:** Timothy G. Barraclough, Samraat Pawar.

**Visualization:** Dimitrios—Georgios Kontopoulos.

**Writing – original draft:** Dimitrios—Georgios Kontopoulos, Thomas P. Smith, Timothy G. Barraclough, Samraat Pawar.

**Writing – review & editing:** Dimitrios—Georgios Kontopoulos, Thomas P. Smith, Timothy G. Barraclough, Samraat Pawar.

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
