## [Editor Report · Decision Letter 0]

9 Jan 2020

Dear Dr Kontopoulos, 

Thank you for submitting your manuscript entitled "Adaptive evolution shapes the present-day distribution of the thermal sensitivity of population growth rate" for consideration as a Research Article by PLOS Biology.

Your manuscript has now been evaluated by the PLOS Biology editorial staff and I am writing to let you know that we would like to send your submission out for external peer review.

Please re-submit your manuscript within two working days, i.e. by Jan 11 2020 11:59PM.

Kind regards,

Lauren A Richardson, Ph.D

Senior Editor

PLOS Biology

---

## [Decision Letter · Decision Letter 1]

4 Feb 2020

Dear Dr Kontopoulos,

Thank you very much for submitting your manuscript "Adaptive evolution shapes the present-day distribution of the thermal sensitivity of population growth rate" for consideration as a Research Article at PLOS Biology. Your manuscript has been evaluated by the PLOS Biology editors, an Academic Editor with relevant expertise, and by several independent reviewers.

As you will read, the reviewers appreciated many aspects of your study. However, Reviewer #3 raises a number of key issues with the phylogenetic methods employed. Further analyses are needed in a revision to support these conclusions.

In light of the reviews (below), we will not be able to accept the current version of the manuscript, but we would welcome re-submission of a much-revised version that takes into account the reviewers' comments. We cannot make any decision about publication until we have seen the revised manuscript and your response to the reviewers' comments. Your revised manuscript is also likely to be sent for further evaluation by the reviewers.

We expect to receive your revised manuscript within 2 months. 

**IMPORTANT - SUBMITTING YOUR REVISION**

*Re-submission Checklist*

*Published Peer Review*

*PLOS Data Policy*

*Blot and Gel Data Policy*

Sincerely,

Lauren A Richardson, Ph.D

Senior Editor

PLOS Biology

REVIEWS:

Reviewer #1: 

The research presented here aims to answer two questions: How variation in thermal sensitive accumulates across multiple autotroph and heterotroph groups (meaning, over time?), and then, whether variation in thermal sensitivity is shaped by environmental selection. This is an important question and of great interest to scientists across disciplines of biology, from physiology to ecology, and the results here are certainly novel and exciting. Variation among thermal sensitivities among taxa has certainly been demonstrated before, and so has variation in temperature dependence of metabolic rates among prokaryote and eukaryote taxonomic groups (e.g., White et al Proc Roy Soc 2012 279: 1742, Galmes et al, Photosynthesis research 2015: 123(2) 183-201.) But, a robust analysis explicitly testing hypotheses from MTE about the universality of activation energy parameters that quantitatively considers phylogenies lacking, to my knowledge. Therefore, I believe this study constitutes a substantial and important contribution to the literature. My belief would be confirmed if the authors could explicitly relate their findings to the two studies noted above (and others if I've missed them) that reach similar findings by different methods (White et al) or different findings (Galmes et al) than reported in the present study.

I should note that I do not have expertise in phylogenetic methods or analyses, and so I cannot evaluate the authors' phylogenetic approach, assumptions or interpretations here. These methods are central to their study and inferences, so I feel uncomfortable providing a strong recommendation on publication or not. I hope there are other reviewers with expertise in phylogeny to provide the editor with a full assessment of this paper.

My main comments are meant to be constructive to the authors in revising their paper to be more clear to readers not expert in this topic. As I mentioned already, I think the findings of this paper are of general interest, but the potential limitations or assumptions in the work are not clear to a non-specialist reader as the manuscript is currently written. I recommend making much more clear and explicit the set of hypotheses tested, and the assumptions inherent in the hypothesis as well as in the methods used to test them. I recommend revising the introduction to explicitly state a set of testable (rejectable) hypotheses. I find myself trying to figure out what the hypotheses are, but really it is the job of the authors to tell me. My reading of the introduction leads me to think that the hypotheses being tested here are: 

H0: there is no phylogenetic signal to thermal sensitivity traits (as defined by the empirically fit sharpe-schoolfield equation). This would be consistent with MTE's UTD in its simplest form. (The authors refer to this as an MTE assumption - line 89.)

One prediction associated with this hypothesis that the authors have given is that measurement error could explain any patterns in the data, rather than phylogenetic signal. The introduction nicely explains what patterns would support or refute this prediction. I would like to see the alternative predictions introduced as clearly.

Another hypothesis I draw out of the introduction is: 

H1: there is phylogenetic signal to thermal sensitivity traits, reflecting environmental conditions experienced by species or groups during their evolution. The authors could be more specific about what these patterns might be expected to be. 

On the geographic variation hypotheses (H2, perhaps), there are alternate explanations that should be considered (especially because thermal variation (annual?) as a predictor was not explicitly tested or even shown. Furthermore, rationale should be given for how (annual?) thermal variation is expected to matter for organisms with such high rates of population growth and presumably evolution. Additionally, many plankton have resting stages in which their populations do not grow during harsh conditions. So the link between thermal variation and diversity of thermal traits in unicellular algae needs to be explained a bit more, and alternative need to be explored. Many ecological and evolutionary patterns and processes vary with latitude, other than thermal variation. Most notably, biodiversity and evolutionary history (tropical groups are generally older). These should be considered or at least mentioned. When this comes up in the discussion (line 279), one alternative explanation is considered (MCA). The finding that E varies with latitude but not W may (or may not) be consisted with explanations other than simply different sample sizes.

The first paragraph of the discussion references two hypotheses, and it would be stronger to see these set up more obviously in the introduction. It is there, but the writing in that first paragraph of the discussion is much more clear in terms of presenting the hypotheses tested.

Lines 184-186: 

"This high convergence in thermal sensitivity space by diverse lineages suggests that variation in the two TPC parameters is mainly driven by adaptation to local environmental conditions, irrespective of the species evolutionary history". Here is a place where the strength of inference and findings would be more clear if a (full?) set of alternate hypotheses were laid out clearly. From what is written, I infer that the possible interpretations for patterns of variation in TPC parameters across species are: 

- Evolutionary history (presumably independent of the environmental influence?)

- Environmental conditions (independent of evolutionary history, as noted by the authors)

- Measurement error (reject above)

- What are other possibilities - constraints on possible traits associated with physical processes (rather than genetic constraints associated with evolution)? Can alternatives be rejected? Is an interaction between evolutionary history and environment possible, and if so, how would that be diagnosed?

Line 204 - 206:

This seems like an interesting result but needs more explanation and setup. Again, a set of hypotheses or predictions might help here. As a reader, I was unprepared for the possible outcomes of this test, and therefore it was unclear what were the implications of their actual finding. It would also help if, when introducing possible outcomes, the authors could highlight key assumptions of their methods and whether those assumptions might lend uncertainty to the conclusions. I think this would help a general biologist reader of PLoS biology who does not have expertise in phylogenetics (like myself).

The results and analysis in Figure 6 are hard to understand. It is not clear why variation in thermal sensitivity among taxa increases with time, nor how one can confidently reconstruct historical thermal sensitivities having previously concluded that they are not explained by evolutionary history but rather by environmental conditions. Does the reconstruction in figure 6 not assume that these traits are affected by evolution, and therefore contradict the previous result?

Methods: 

Though I can't evaluate the phylogenetic methods, I do think the authors' general statistical approach of ensuring that their datasets do not include overlapping species, and the consideration of possible covariation among TPC parameters in their analysis are appropriate. 

How much confidence do the authors have in the phylogenetic estimates of time (their relative time axis)? These uncertainties should be explained. Not being an expert, I assume it is likely that evolutionary relationships are better resolved for more recent divisions compared to those in deeper time; could the uncertainty in phylogenies associated with time explain part of the results in figure 3?

Minor comments: 

There seems to be increasing evidence that the E value of photosynthesis may not universally be 0.32, as was suggested by Allen et al 2005. There has been support for this value, mostly in algae, but less in terrestrial plants. You might consider removing this sentence (line 29-31). However, this is very much unresolved. The work I've seen that convinced me that 0.32 is not well-supported is not yet published! So I simply suggest this edit; it's up to you to leave it or keep it.

Another point that would increase clarity for a general reader would be to use the terms rather than the parameter names (or, in addition to). For example, on line 210, the sentence could refer to temperature dependence and thermal niche breadth rather than just E and W0.

--------------

Reviewer #2: 

Review of "Adaptive evolution shapes the present-day distribution of the thermal sensitivity of population growth rate"

In this article, the authors undertake several analyses to determine whether features of thermal performance curves (TPCs) in phytoplankton and prokaryotes have/can evolve or are constrained by underlying thermodynamic constraints of biochemistry. Altogether, the authors conclude that, contrary to the hypothesis that there is universal temperature dependence (UTD), that activation energy (E, describing the increase in the left side of the TPC) has diverged through time and that different clades explore considerable variation in E within clades. They also show that much of this variation has occurred in more recent parts of their timelines. They make similar conclusions that TPC breadth (Wop) displays these same patterns, which are also consistent with the evolution of respiration and photosynthesis in plants, which can be considered underlying drivers of growth. Finally, they find that E declines with latitude for all types of TPCs in their analysis, but not so for Wop. 

OK, there is a lot going on in this article, so sorry it took a while. Overall, I think it's a great contribution. It provides fundamental insights into the evolution of thermal tolerance, not just retrospectively through the lens of phylogeny but also that further evolution of thermal sensitivity is possible. It's a novel take on a core problem. This paper should be the final death knell of the UTD as we currently think of it, not that there couldn't be other ways that thermodynamics constrain physiology. The paper is thorough, well-written, and the figures are engaging and informative. 

I do have some questions and comments, a few little things to annoy the authors with, but hopefully they will be useful. 

First, since these are most if not all (presumably) lab cultures, to what extent is adaptation/acclimation to a typical lab temperature influencing the perceived pattern of evolution? If over many generations the cultures have shifted their TPCs to adapt to ~22C, as is clearly possible given (Padfield et al. 2016), then would the results reflect some convergence around the lab environment rather than the phylogenetic pathways that these organisms travel in the wild? I don't believe there is anything to be done about this possibility in the analysis, but it could be influencing the results.

Abstract. Where it says (there are no line numbers here) 'typically-experienced thermal range', I would change this, because there is no clear reason why the rising portion of a curve is where most of the experienced temperatures are. Certainly many species can experience typical temperatures around the peak. Also see line 22 in the introduction. I would also say your results are 'consistent with' rather than 'driven by two opposing…' factors. Only if you could isolate the effect of temperature fluctuations explicitly (e.g., with experiments) could you go as far as 'driven'. Also here, evolution can occur at ecological time scales for anything; this is not a unique feature of microbes.

Line 17 - I don't think this is well-established, although I agree some folks would like it to be. See (Darveau et al. 2002).

Line 45 - I think you are overstating the restrictiveness of the UTD. Even Gillooly's original paper (Gillooly et al. 2001) stated a range of eV as "0.2 and 1.2 eV".

Line 46 - I don't think we really expect a Gaussian distribution for a measurement that cannot be negative, nor is 'very substantial' measurement error needed to generate that right skew. From my experience, even bootstrapped estimates of positive parameters are generally right-skewed without bunching up at 0. I don't think this takes away from the value of your work, but the argument seems to be a bit of a straw man. See also line 354. Maybe it's a matter of degrees.

Line 63 - Seems like you need some references here. How about (Alexander Jr and McMahon 2004).

Line 72 - Why does this require 'frequent convergence'? Seems like quickly adapting would be sufficient to break down the ability to detect heritability.

Line 141 - If you are doing this analysis phylogenetically, why are you separating the datasets at all? Can't you just run it all together?

Lines 165-168 - As I understand this here, as well as described by (Harmon et al. 2003), relative disparity varies from 0 to 1, as the subclade disparity is divided by the across-clade disparity. As described by Harmon et al, "Values near 0 imply that subclades contain relatively little of the variation present within the taxon as a whole and that, consequently, most variation is partitioned as among-subclade differences; conversely, values near 1 imply that subclades contain a substantial proportion of the total variation and thus are likely to overlap extensively". This seems to be in line with your description, but my question is that Figure 3 shows a lot of disparity above 1. Can you clarify why this can happen or confirm that there is not an error, or, clarify what I might be missing?

Line 172 - 'Exact' might be a stretch, given that the phylogenies and nodes themselves are hypotheses and estimates.

Line 176 - A bit counterintuitive here. Why is adaptive radiation characterized by decelerating evolution? Seems to me like examples of adaptive radiation are generally characterized by fast evolution accompanied by key innovations, which may or may not be part of the disparity. Why not just delete phrase after 'space'?

Line 211 - What does subtree refer to here?

Line 214 - I think optimum is the wrong word for the UTD of 0.65. As indicated in the same sentence, the value is thermodynamically-constrained, which is a biochemical/physical property and may or may not be an optimum. Likewise in line 218 and later, I would suggest a 'global mean set by thermodynamic constraints' not a 'global optimum'.

Line 261 - Might want to specify that it's a thorough analysis in prokaryotes and phytoplankton, since otherwise it suggests you have all kinds of species. 

Line 267 - 'are' not 'is' here, I see two subjects.

Line 268 - Suggest here adding 'rate-limiting enzyme … of respiration'.

Line 285-286 - Not sure this argument makes sense. E doesn't really tell you breadth on its own. A low E doesn't automatically indicate narrow or wide, because the breadth is determined by the distances between upper and lower critical values and Tpk that could be connected by steep or shallow rises. I would also suggest that Wop is an arbitrary indicator of width making no assumptions about realized environmental temperatures. You might be best off getting rid of this statement, or at least getting clearer on the message.

Line 310 - Tpk in respiration, the purported underlying driver of rmax, is negatively associated with latitude (DeLong et al. 2018), which I think complicates this explanation a bit, since latitude is invoked as a proxy for fluctuations.

Line 320 - See (Uiterwaal et al. n.d.) for a recent example of a link between TPC evolution and biotic interactions as well as (Luhring and DeLong 2016) for direct evidence that biotic interactions alter rmax TPCs.

Line 348 - Not sure I see the mechanistic explanation. Even the previous paragraph explained a range of mechanisms that could underlie the changes in TPCs. Seems to me you've demonstrated patterns of evolution that clearly contradict the UTD but haven't explained the mechanisms. 

Line 358-362 - I don't get this. I don't think the disparity analysis is exactly evidence of convergence, which would require demonstrating that different lineages evolved the same trait in the same conditions. It says that the clades strongly overlap in state space, which may or may not contain cases of convergence.

Signed, John P. DeLong

References

Alexander Jr, J. E., and R. F. McMahon. 2004. Respiratory response to temperature and hypoxia in the zebra mussel Dreissena polymorpha. Comparative Biochemistry and Physiology Part A: Molecular & Integrative Physiology 137:425-434.

Darveau, C.-A., R. K. Suarez, R. D. Andrews, and P. W. Hochachka. 2002. Allometric cascade as a unifying principle of body mass effects on metabolism. Nature 417:166-170.

DeLong, J. P., G. Bachman, J. P. Gibert, T. M. Luhring, K. L. Montooth, A. Neyer, and B. Reed. 2018. Habitat, latitude and body mass influence the temperature dependence of metabolic rate. Biology Letters 14:20180442.

Gillooly, J. F., J. H. Brown, G. B. West, V. M. Savage, and E. L. Charnov. 2001. Effects of size and temperature on metabolic rate. Science 293:2248-2251.

Harmon, L. J., J. A. Schulte, A. Larson, and J. B. Losos. 2003. Tempo and Mode of Evolutionary Radiation in Iguanian Lizards. Science 301:961-964.

Luhring, T. M., and J. P. DeLong. 2016. Predation changes the shape of thermal performance curves for population growth rate. Current Zoology 62:501-505.

Padfield, D., G. Yvon-Durocher, A. Buckling, S. Jennings, and G. Yvon-Durocher. 2016. Rapid evolution of metabolic traits explains thermal adaptation in phytoplankton. Ecology Letters 19:133-142.

Uiterwaal, S. F., I. T. Lagerstrom, T. M. Luhring, M. E. Salsbery, and J. P. DeLong. (n.d.). Trade-offs between morphology and thermal niches mediate adaptation in response to competing selective pressures. Ecology and Evolution n/a.

--------------

Reviewer #3: 

This study seeks to understand how thermal sensitivity of three key traits among prokaryotes, phytoplankton, and plants evolve along the branches of their respective phylogenetic trees. This is potentially important for knowing how species will respond to human-mediated climate change. The manuscript reports that thermal sensitivity does not evolve in a gradual manner (i.e. via random walk) but can differ considerably even between closely related species, and that there is a weak negative association between thermal sensitivity and latitude.

The paper is very nicely written - its introduction was a pleasure to read. However, in my opinion, the analyses and methods leave a great deal to be desired. The analyses are poorly described and highly unconventional (which is not a bad thing but in this case seems totally unnecessary and confused). It is not clear at all how the analyses relate to the hypotheses under investigation nor is it clear how various comparative tests should be interpreted and reconciled with each other.

I think there are some basic problems with the way the authors describe their null model - they often talk about traits evolving randomly; however, it is more accurate to say by random walk or Brownian motion. And even if a trait is evolving by random walk along the branches it does not rule out that the trait has evolved under selection - though this is a relatively minor problem. What is more serious is that the set of three analyses basically describe the same phenomenological pattern in the data but have distinct interpretations - thus the reader is left wondering how to interpret the results. 

The first MCMCGLMM analyses takes all traits together and estimates heritability (h2). It is not clear how the authors constructed these multi-response models to achieve this and the performance of such models has never been assessed. Theoretically they should give identical results to more conventional tests such as Pagel's Lambda - but the prior on the random effects in the MCMCGLMM analyses can be influential. I think it would be better for the authors to conduct a more reliable test using Lambda (with or without estimating covariances among traits - depending on significance given the tree), or at least such a test should be compared with the MCMCGLMM result. In any case if we take the analyses at face value then it shows that there is some phylogenetic signal in the data, but it is not perfect - i.e. h2 is lower than one. This could be an honest indication that there is lack of signal in the trait or it could be noise in the data. The authors attempt to show that the signal is not owing to noise, but it is not clearly explained and was not convincing to me in its current state. 

They then go on to try to determine if there is rate variation along and among the branches of the trees. To do this they use the 'Free model' and the 'Stable model'. These are strange choices as they don't identify significant shifts in the rate of evolution (whereas several other methods do, see below). The Free model is heavily overparameterized and at best should only be used to provide a rough visualization of rate variation (see Mooers et al, 1999 and Thomas and Freckleton, 2010 - cited in the manuscript). The Stable model was presented to estimate ancestral states rather than rates (although these are two sides of the same coin). Neither method formally identifies areas of the tree (or branches) which are evolving at a significantly different rate, yet this what the authors need to support their conclusion. Methods such as those described in Rabosky (2014, PLoS ONE 9(2): e89543), Landis et al (Systematic Biology 62(2): 193-204), Venditti et al (2011, Nature 479(7373): 393-396), Eastman et al (2011, Evolution 65(12): 3578-3589) can be used to achieve this - this list is not exhaustive . Looking at the results of the 'Free model' and the 'Stable model', most of the rate variation is at the tips of the tree which is what one would expect give the lack of phylogenetic signal in their previous analyses. It does not in itself support the idea that there are intense episodes of adaptation with closely related species having very distinct rates, which is their main conclusion.

The authors' third analyses seem superfluous. Here they estimate a model with an evolutionary optimum - this will again result in a pattern that one would expect if there was a lack of phylogenetic signal in the data. It is not clear what these analyses add to the paper (other than attractive figures) or how they should be interpreted. 

There is no attempt to discriminate between the results of these first three sets of analyses which have very different interpretations despite similarities in patterns. I think the authors need to take a step back and think about how the statistical test they are carrying out relates to the hypotheses. I believe their question can be addressed by a single set of analyses to identify significant shifts in the rate of evolution on the tree using the methods I mentioned previously.

The last set of analyses regarding the latitude is also very confusing and unnecessarily complex. There is no need to estimate models with and without the variance-covariance matrix as a random effect. If there is no signal in the data, h2 will be very low - thus the variance explained by the random effect will be negligible and will reduce to be equivalent to the non-phylogenetic test.

Owing to the reasons outlined above, while I find the subject matter interesting and potentially suitable for PLoS Biology, I can't support its publication in the current form. In addition, based on the analyses provided I cannot predict whether the result will be of interest to the readership of PLoS Biology.

---

## [Decision Letter · Decision Letter 2]

11 May 2020

Dear Dr Kontopoulos,

Thank you very much for submitting a revised version of your manuscript "Adaptive evolution shapes the present-day distribution of the thermal sensitivity of population growth rate" for consideration as a Research Article at PLOS Biology. This revised version of your manuscript has been evaluated by the PLOS Biology editors, the Academic Editor and two of the original reviewers.

In light of the reviews (below), we offer you the opportunity to address the remaining points from the reviewers in a revised version that we anticipate should not take you very long. We will then assess your revised manuscript and your response to the reviewers' comments and we may consult the reviewers again. I should warn you, however, that we will only consult the reviewers one more time, and if they remain unsatisfied then we will not consider the paper further.

We expect to receive your revised manuscript within 1 month.

**IMPORTANT - SUBMITTING YOUR REVISION**

*Resubmission Checklist*

*Published Peer Review*

*PLOS Data Policy*

*Blot and Gel Data Policy*

Sincerely,

Roli Roberts

Senior Editor

PLOS Biology

REVIEWERS' COMMENTS:

Reviewer #2:

[and see attached figure]

Review of "Adaptive evolution shapes the present-day distribution of the thermal sensitivity of population growth rate"

I previously reviewed this article. Overall, I think the paper is improved and the responses were reasonable. I just have a few areas that I still find challenging.

1) The disparity levels going above 1 still seems confusing. The authors write "The disparity-through-time method compares i) the mean disparity within subclades with ii) the total disparity across the phylogeny. Therefore, it is mathematically possible to get values much greater than 1 when the trait variance within subclades (at a given time point) is much greater than the trait variance across the tree." What I don't understand is how there can be more variance within than across subclades, when the variation across subclades would seem to me to be inclusive of all the variation occurring within subclades. I suspect there is some crucial detail missing.

2) The use of Wop and the link between E and breadth is still confusing. There are two issues. 

2A) First, choosing Wop as you have done does not require an assumption. It is just a choice. I think what you might be trying to say is that you chose this specific metric (instead of, say, the difference between the upper and lower critical temperatures) because you feel this metric best captures the curve's breadth in the typical range of experienced temperatures. 

2B) Second, the explanation that E reflects breadth still doesn't make sense to me. The authors write "an increase in Wop (a larger operational niche width) will necessarily be associated with a shallower rising slope (a low E value)." Yet it is easy to show that this is not true. Consider the attached cartoon. Two curves, same Tpk. The lower curve has a lower E and a smaller Wop - opposite of the 'necessary' pattern. So I find this confusing, and it influences interpretation of the results as well. For example, the explanation for Figure S4 is that it has to be that way. Does it? I think, perhaps, the authors are making additional unstated assumptions about correlations among parameters, such that the proposed necessary correlation will arise under particular scenarios, such as, for example, Tpk and E are negatively correlated.

3) It is still not very clear why the disparity analysis indicates convergent evolution. The authors clarify that there are different definitions of convergence that focus on pattern or process, and that they 'use the term "convergent evolution" to refer to remote clades that increasingly overlap in the parameter space due to independent evolution of similar thermal sensitivity values.' Note the point here is 'similar thermal sensitivity values'. To me the disparity results seem to say that the subclades have evolved similar 'ranges' - that is, a lot of overlap in extent - rather than showing convergence toward similar values within that extent. The pattern appears to be (and this is a major point of the paper) that lineages are exploring a lot of thermal parameter space. Overlapping a lot because multiple lineages have evolved a wide range of parameter values does not seem to me to be consistent with the idea of convergent evolution.

A few more minor things:

Figure 1 legend - "E and ED (eV) control how smoothly". Isn't it how steeply, since a low E does not cause a 'rough' rise?

L123 - Evolution would not favor species but individuals with lower thermal sensitivities.

L214 - The 'true' disparity is not really knowable. The 'estimated' or 'observed' disparity is - 'observed' is the word you used in the F3 legend. I would use that.

Reviewer #3:

I will speak to the author's comments using the numbers they include in their rebuttal:

30) I think the removal of the notion of gradual evolution has improved the manuscript.

31) The authors are misguided in their assessment: the MCMCglmm analyses are not far superior. h2 and Lambda are mathematically identical (if for lambda you take a mean of the within-species values and estimate co-variances among traits) - this is clear in the original paper describing heritability in the context of phylogenetic analysis, which the authors cite for a description of the model! My specific worry is that in the MCMCglmm implementation, h2 can be very sensitive to the prior. (Incidentally, what prior was on the random effect here, and what does the distribution look like if you turn the likelihood function off and just target the prior?) This sensitivity to prior choice is not the case for Lambda, owing to the fact that it is implemented within the more conventional GLS framework. I therefore still maintain that given this, it is important to check the test using the more well-established method. Estimating co-variances and Lambda can be done in many GLS packages including BayesTraits - Pagel's own program. Incidentally, phylogenetic imputation can also be done in a GLS framework again in number of packages including BayesTraits. 

32) Given that the Levy process analyses have been conducted and are by far better justified I believe they should be reported in the main text - the others can be removed altogether as they are no longer needed.

33) I still can't see the value of these analyses if authors believe the previous analyses - they all show the same thing - but the interpretations are not consistent. I believe they must be removed - how can these results and the Levy process results be reconciled?

34) The reasons I state above mean we cannot believe the h2 until it has been verified by lambda so this point still stands.

---

## [Decision Letter · Decision Letter 3]

18 Jul 2020

Dear Dr Kontopoulos,

Thank you very much for submitting a revised version of your manuscript "Adaptive evolution shapes the present-day distribution of the thermal sensitivity of population growth rate" for consideration as a Research Article at PLOS Biology. This revised version of your manuscript has been evaluated by the PLOS Biology editors, the Academic Editor and one of the original reviewers.

In light of the reviews (below), we are pleased to offer you the opportunity to address the remaining points from reviewer #3 in a revised version that we anticipate should not take you very long.

We note that we had previously said that we would not consider the manuscript further if you failed to satisfy the reviewers; however, because this reviewer remains overall very positive about your study, we are prepared to give you one final chance. Further consideration is absolutely predicated on you addressing this final issue. We will then assess your revised manuscript and your response to the reviewer's comments and we may consult this reviewer again.

We expect to receive your revised manuscript within 1 month.

**IMPORTANT - SUBMITTING YOUR REVISION**

*Resubmission Checklist*

*Published Peer Review*

*PLOS Data Policy*

*Blot and Gel Data Policy*

Sincerely,

Roli Roberts

Senior Editor

PLOS Biology

REVIEWER'S COMMENTS:

Reviewer #3:

I find the difference between Lambda and H2 quite alarming - and I do not really understand why the authors would not use the MCMC option in a Bayesian context for comparison which allows multiple data points per taxa - especially as MCMCglmm also provides a posterior distribution of h2. It would be ideal to see the two distributions together. Such differences in the level of phylogenetic signal have the potential to change the results altogether - I am not sure if that is the case here as I don't have enough information. I think this is a critical point as phylogenetic signal is fundamental to this study. I feel like I am sounding like a broken record here, but the analyses I am suggesting a very easy to do and I can't understand why the authors don't want to perform/show them.

I very much like this paper and hope to see it published on PLoS Biology - but I think this final piece of the puzzle is missing.

---

## [Editor Report · Decision Letter 4]

12 Aug 2020

Dear Dr Kontopoulos,

Thank you for submitting your revised Research Article entitled "Adaptive evolution shapes the present-day distribution of the thermal sensitivity of population growth rate" for publication in PLOS Biology. I have now obtained further advice from reviewer #3 and have discussed their comments with the Academic Editor. 

You'll see that reviewer #3 is now largely satisfied, and the Academic Editor agrees that this reviewer's final point can be addressed textually. Based on the review, we will probably accept this manuscript for publication, assuming that you will modify the manuscript to address the remaining points raised by reviewer #3. Please also make sure to address the data and other policy-related requests noted at the end of this email.

We expect to receive your revised manuscript within two weeks. Your revisions should address the specific points made by each reviewer. In addition to the remaining revisions and before we will be able to formally accept your manuscript and consider it "in press", we also need to ensure that your article conforms to our guidelines. A member of our team will be in touch shortly with a set of requests. As we can't proceed until these requirements are met, your swift response will help prevent delays to publication.

*Copyediting*

*Published Peer Review History*

*Early Version*

*Submitting Your Revision*

Sincerely,

Roli Roberts

Senior Editor,

rroberts@plos.org,

PLOS Biology

DATA POLICY:

We note that you are intending to deposit your phylogeny and parameter estimates in Figshare; please could you do so, include the URL in the manuscript, and provide us with access to it? Please could you also deposit (e.g. in Github) any code used for the study?

In addition, we ask that all individual quantitative observations that underlie the data summarized in the figures and results of your paper be made available in one of the following forms:

Regardless of the method selected, please ensure that you provide the individual numerical values that underlie the summary data displayed in all of the main and Supplementary figure panels as they are essential for readers to assess your analysis and to reproduce it. NOTE: the numerical data provided should include all replicates AND the way in which the plotted mean and errors were derived (it should not present only the mean/average values).

REVIEWER'S COMMENTS:

Reviewer #3:

One can see that the results of the phylogenetic signal are quite different from different methods! As the authors are happy to include this and discuss them I am happy to recommend publication – with one caveat. The authors seem to think that the difference is associated with the way the program deal with missing data. I doubt very much this is the case – one should not be able to get more information about signal from imputed/missing data points! I am sure that the difference between mcmcglmm and Baystraits is the prior. The prior is far more simple to implement in a GLS framework (BayesTraits) compared to the way it has to be implemented in mcmcglmm. I think the authors should at least posit this idea – then the reader can decide and/or test it in the future.

---

## [Editor Report · Decision Letter 5]

14 Sep 2020

Dear Dr Kontopoulos,

On behalf of my colleagues and the Academic Editor, Simon A. Levin, I am pleased to inform you that we will be delighted to publish your Research Article in PLOS Biology. 

Early Version

PRESS 

Kind regards,

Vita Usova, 

Publishing Editor

PLOS Biology

on behalf of

Roland Roberts,

Senior Editor

PLOS Biology